# Uncertainty-Regularized Diffusional Subgoals for Hierarchical Reinforcement Learning

## Abstract

Hierarchical reinforcement learning (HRL) aims to solve complex tasks by making decisions across multiple levels of temporal abstraction. However, off-policy training of hierarchical policies faces non-stationarity issues because the low-level policy is constantly changing, which makes it difficult for the high-level policy that generates subgoals to adapt. In this paper, we propose a conditional diffusion model-based approach for subgoal generation to mitigate these non-stationarity challenges. Specifically, we employ a Gaussian Process (GP) prior on subgoal generation as a surrogate distribution to regularize the diffusion policy and inform the diffusion process about uncertain areas in the action space. We introduce adaptive inducing states to facilitate sparse GP-based subgoal generation, enhancing sample efficiency and promoting better exploration in critical regions of the state space. Building on this framework, we develop an exploration strategy that identifies promising subgoals based on the learned predictive distribution of the diffusional subgoals. Experimental results demonstrate significant improvements in both sample efficiency and performance on challenging continuous control benchmarks compared to prior HRL methods.

## 1 Introduction

In the domain of Hierarchical Reinforcement Learning (HRL), the strategy to simplify complex decision-making processes involves structuring tasks into various levels of temporal and behavioral abstractions. This strategy excels particularly in environments where the challenges include long-term credit assignment and sparse rewards, making it an ideal solution for long-horizon decision-making problems. Among the prevailing HRL paradigms, goal-conditioned HRL has been extensively explored in various studies (Dayan & Hinton, 1992; Schmidhuber & Wahnsiedler, 1993; Kulkarni et al., 2016; Vezhnevets et al., 2017; Nachum et al., 2018; Levy et al., 2019; Zhang et al., 2020; Li et al., 2021; Kim et al., 2021; Li et al., 2023). Within this framework, a high-level policy decomposes the primary goal into a series of subgoals, effectively directing the lower-level policy to achieve these subgoals. The efficacy of this approach hinges on the ability to generate subgoals that are semantically coherent and achievable, providing a strong learning signal for the lower-level policies. The hierarchical structure not only enhances the learning process's efficiency but also considerably improves the policy's overall performance in solving complex tasks.

Nonetheless, the hierarchical structure in HRL has induced non-stationary state transitions, which remains a key challenge in off-policy training of a hierarchy of policies. The crux of the issue lies in the fact that the same high-level action taken under the same state in the past may result in significantly different low-level state transitions. This is caused by the constantly changing low-level policy, which renders the previous experience invalid for training. When all policies within the hierarchy are trained simultaneously, the high-level transition will constantly change as long as the low-level policy continues to be updated. Parallel learning of hierarchical policies remains feasible, but only if the high-level policy can efficiently adapt itself to the updated low-level policy.

Previous works like HIRO (Nachum et al., 2018) and HAC (Levy et al., 2019) have attempted to address this non-stationarity problem through a relabeling strategy that utilizes hindsight experience replay (HER) (Andrychowicz et al., 2017a). This involves relabeling past experiences with high-level actions, *i.e.*, subgoals, that maximize the probability of the past lower-level actions. Essentially, the subgoal that induced a low-level behavior in the past experience is relabeled so that it potentially

induce the similar low-level behavior with the current low-level policy. However, the relabeling approach does not facilitate efficient training of the high-level policy to comply promptly with updates to the low-level policy. It consistently generates incompatible subgoals, further exacerbating the non-stationarity issue. Consequently, such unfit state transitions in off-policy training lead to improper learning of the high-level value function, negatively affecting high-level policy exploration.

In this work, we propose a diffusion model (Sohl-Dickstein et al., 2015; Song et al., 2021a; Ho et al., 2020) based approach for subgoal generation, aiming to address the non-stationarity issue in HRL. Our approach involves modeling the state-conditional distribution of subgoals at the high level, harnessing a Gaussian Process (GP) prior as an explicit surrogate distribution for subgoals. The GP prior potentially reduces the amount of training data needed for the diffusion model to converge, and more importantly it could help inform the diffusion policy about state regions that are more uncertain when learning the conditional distribution by providing an uncertainty quantification. To improve the sample efficiency and exploration of the high-level policy, we introduce *inducing states*, *i.e.*, learnable pseudo states, which adaptively summarize the states that are informative to explore. These learnable pseudo states facilitate finer modeling in critical regions of the state space which pushes for better exploration. Once learned, the inducing states provides implicit subgoals, which are used to predict the distribution of subgoals generated by the high-level policy. Building on our GP-based probabilistic formulation, we propose an exploration strategy that explicitly identifies promising subgoals based on the learned surrogate distribution of diffusional subgoals. Our key insight is that novel subgoals should be both reachable and capable of guiding the agent toward unexplored areas, consistent with the probabilistic nature of the predictive distribution of subgoals. This approach is more efficient and stable than relying on visit counts for subgoals such as Li et al. (2022). Instead of focusing on visit coverage, our method proposes reasonable subgoals based on the non-parametric distribution, which can be learned more data-efficiently during training.

Our main contributions are summarized as follows:

- We introduce conditional diffusion modeling approach for subgoal generation to address the non-stationarity issue of off-policy HRL.

- We employ a Gaussian Process (GP) prior to model the state-to-subgoal mapping, providing a probabilistic framework that quantifies uncertainties across the state space, regularizes the diffusional subgoal generation, and implicitly informs exploration in uncertain areas.

- We propose an inducing states informed exploration strategy that seeks promising subgoals based on the learned predictive distribution of diffusional policy.

## 2 PRELIMINARIES

**Goal-conditioned HRL**  In reinforcement learning (RL), agent-environment interactions are modeled as a Markov Decision Process (MDP) denoted by $M = \langle \mathcal{S}, \mathcal{A}, \mathcal{P}, \mathcal{R}, \gamma \rangle$, where $\mathcal{S}$ represents the state space, $\mathcal{A}$ denotes the action set, $\mathcal{P} : \mathcal{S} \times \mathcal{A} \times \mathcal{S} \to [0, 1]$ is the state transition probability function, $\mathcal{R} : \mathcal{S} \times \mathcal{A} \to \mathbb{R}$ is the reward function, and $\gamma \in [0, 1)$ signifies the discount factor. A stochastic policy $\pi(a|s)$ maps any given state $s$ to a probability distribution over the action space, with the goal of maximizing the expected cumulative discounted reward $\mathbb{E}_\pi[\sum_{t=0}^{\infty} \gamma^t r_t]$, where $r_t$ is the reward received at discrete time step $t$.

In a continuous control RL setting, modeled as a finite-horizon, goal-conditioned MDP $M = \langle \mathcal{S}, \mathcal{G}, \mathcal{A}, \mathcal{P}, \mathcal{R}, \gamma \rangle$, where $\mathcal{G}$ represents a set of goals, we employ a Hierarchical Reinforcement Learning (HRL) framework comprising two layers of policy akin to Nachum et al. (2018). This framework includes a high-level policy $\pi_h(g|s)$ that generates a high-level action, or subgoal, $g_t \sim \pi_h(\cdot|s_t) \in \mathcal{G}$, every $k$ time steps when $t \equiv 0 \pmod{k}$. Between these intervals, a predefined goal transition function $g_t = f(g_{t-1}, s_{t-1}, s_t)$ is applied when $t \not\equiv 0 \pmod{k}$. The high-level policy influences the low-level policy through intrinsic rewards for achieving these subgoals. Following prior work (Andrychowicz et al., 2017a; Nachum et al., 2018; Zhang et al., 2020; Kim et al., 2021), we define the goal set $\mathcal{G}$ as a subset of the state space, *i.e.*, $\mathcal{G} \subset \mathcal{S}$, and the goal transition function as $f(g_{t-1}, s_{t-1}, s_t) = s_{t-1} + g_{t-1} - s_t$. The objective of the high-level policy is to maximize the extrinsic reward $r_{kt}^h$ as defined by:

$$r_t^h = \sum_{i=t}^{t+k-1} R_i, \quad t = 0, 1, 2, \ldots \tag{1}$$

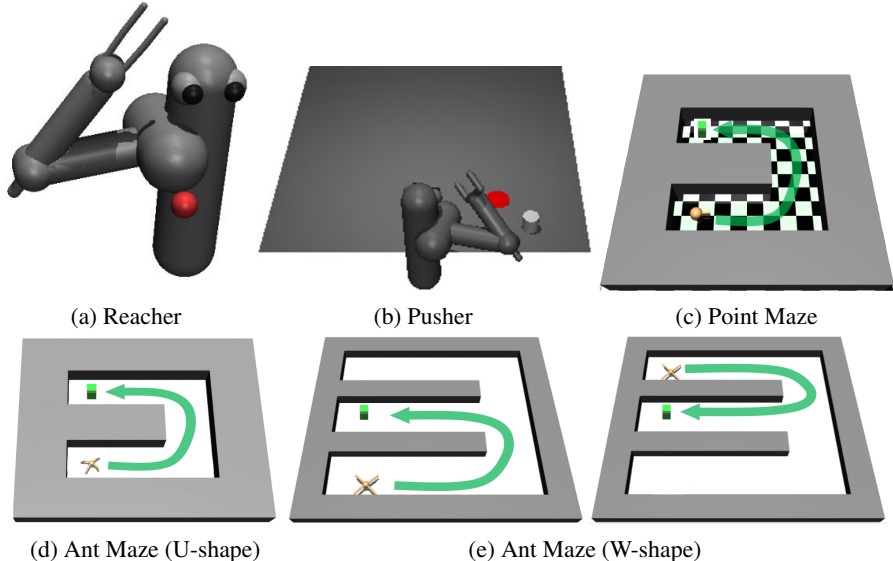

(a) Reacher          (b) Pusher          (c) Point Maze

(d) Ant Maze (U-shape)          (e) Ant Maze (W-shape)

Figure 1: Environments used in our experiments.

where $R_i$ is the environmental reward.

The low-level policy aims to maximize the intrinsic reward granted by the high-level policy. It accepts the high-level action or subgoal $g$ as input, interacting with the environment by selecting an action $a_t \sim \pi_l(\cdot|s_t, g_t) \in \mathcal{A}$ at each time step. An intrinsic reward function, $r_t^l = -\|s_t + g_t - s_{t+1}\|_2$, evaluates the performance in reaching the subgoal $g$.

This goal-conditioned HRL framework facilitates early learning signals for the low-level policy even before a proficient goal-reaching capability is developed, enabling concurrent end-to-end training of both high and low-level policies. However, off-policy training within this HRL framework encounters the non-stationarity problem as highlighted in Section 1. HIRO (Nachum et al., 2018) addresses this by relabeling high-level transitions $(s_t, g_t, \sum_{i=t}^{t+k-1} R_i, s_{t+k})$ with an alternate subgoal $\tilde{g}_t$ to enhance the likelihood of the observed low-level action sequence under the current low-level policy, by maximizing $\pi_l(a_{t:t+k-1}|s_{t:t+k-1}, \tilde{g}_{t:t+k-1})$.

The relabeled subgoals are generally considered to be sampled from a distribution that asymptotically approximates an optimal high-level policy within a stationary data distribution (Zhang et al., 2020; Wang et al., 2023). This leads to the conjecture that learning a conditional distribution based on the relabeled subgoals inherently facilitates the achievement of stationarity in the high-level policy.

**Diffusion Model**    Diffusion models (Sohl-Dickstein et al., 2015; Song et al., 2021a; Ho et al., 2020) have emerged as a powerful framework for generating complex data distributions. These models pose the data-generating process as an iterative denoising procedure $p_\theta(\mathbf{x}_{t-1}|\mathbf{x}_t)$. This denoising is the reverse of a diffusion or forward process which maps a data example $\mathbf{x}_0 \sim q(\mathbf{x}_0)$ through a series of intermediate variables $\mathbf{x}_{1:T}$ in $T$ steps with a pre-defined variance schedule $\beta_i$, according to

$$q(\mathbf{x}_{1:T}|\mathbf{x}_0) := \prod_{t=1}^{T} q(\mathbf{x}_t|\mathbf{x}_{t-1}), \quad q(\mathbf{x}_t|\mathbf{x}_{t-1}) := \mathcal{N}(\mathbf{x}_t; \sqrt{1-\beta_t}\mathbf{x}_{t-1}, \beta_t\mathbf{I}). \quad (2)$$

The reverse process is constructed as $p_\theta(\mathbf{x}_{0:T}) := \mathcal{N}(\mathbf{x}_T; 0, \mathbf{I})\prod_{t=1}^{T} p_\theta(\mathbf{x}_{t-1}|\mathbf{x}_t)$. Parameters $\theta$ are optimized by maximizing a variational bound on the log likelihood of the reverse process $\mathcal{L}_{\text{ELBO}} = \mathbb{E}_{q(\mathbf{x}_{1:T}|\mathbf{x}_0)}[\log \frac{p_\theta(\mathbf{x}_{1:T}|\mathbf{x}_0)}{q(\mathbf{x}_{1:T}|\mathbf{x}_0)}]$.

In this paper, we employ two distinct notations of timesteps: one for the diffusion process and another for the reinforcement learning trajectory. Specifically, we denote the diffusion timesteps with superscripts $i \in \{1, \ldots, N\}$, and the trajectory timesteps with subscripts $t \in \{1, \ldots, T\}$.

## 3 PROPOSED METHOD

In this section, we present our method — **HI**erarchical RL subgoal generation with **DI**ffusion model (HIDI), which explicitly models the state-conditional distribution of subgoals at the higher level. By combining with the temporal difference learning objective, the high-level policy promptly adapts to generating subgoals following a data distribution compatible with the current low-level policy.

### 3.1 DIFFUSIONAL SUBGOALS

We formulate the high-level policy as the reverse process of a conditional diffusion model as

$$\pi_{\theta_h}^h(\boldsymbol{g}|\boldsymbol{s}) := p_{\theta_h}(\boldsymbol{g}^{0:N}|\boldsymbol{s}) = \mathcal{N}(\boldsymbol{g}^N; \boldsymbol{0}, \boldsymbol{I}) \prod_{i=1}^{N} p_{\theta_h}(\boldsymbol{g}^{i-1}|\boldsymbol{g}^i, \boldsymbol{s}), \tag{3}$$

with the end sample $\boldsymbol{g}_i$ being the generated subgoal. Following Ho et al. (2020), a Gaussian distribution $N(\boldsymbol{g}^{i-1}; \mu_{\theta_h}(\boldsymbol{g}^i, \boldsymbol{s}, i), \Sigma_{\theta_h}(\boldsymbol{g}^i, \boldsymbol{s}, i))$ is used to model $p_{\theta_h}(\boldsymbol{g}^{i-1}|\boldsymbol{g}^i, \boldsymbol{s})$, which is parameterized as a noise prediction model with learnable mean

$$\boldsymbol{\mu}_{\theta_h}\left(\boldsymbol{g}^i, \boldsymbol{s}, i\right) = \frac{1}{\sqrt{\alpha_i}}\left(\boldsymbol{g}^i - \frac{\beta_i}{\sqrt{1-\bar{\alpha}_i}}\boldsymbol{\epsilon}_{\theta_h}\left(\boldsymbol{g}^i, \boldsymbol{s}, i\right)\right)$$

and fixed covariance matrix $\boldsymbol{\Sigma}_{\theta_h}\left(\boldsymbol{g}^i, \boldsymbol{s}, i\right) = \beta_i \boldsymbol{I}$.

Starting with Gaussian noise, subgoals are then iteratively generated through a series of reverse denoising steps by the noise prediction model parameterized by $\theta_h$ as

$$\boldsymbol{g}^{i-1} = \frac{1}{\sqrt{\alpha_i}}\left(\boldsymbol{g}^i - \frac{\beta_i}{1-\bar{\alpha}_i}\boldsymbol{\epsilon}_{(\theta_h)}\left(\boldsymbol{g}^i, s, i\right)\right) + \sqrt{\beta_i}\boldsymbol{\epsilon}, \ \boldsymbol{\epsilon} \sim \mathcal{N}(\boldsymbol{0}, \boldsymbol{I}) \text{ if } i > 1, \text{else } \boldsymbol{\epsilon} = 0. \tag{4}$$

The learning objective of our subgoal generation model comprises three terms, *i.e.*, diffusion objective $\mathcal{L}_{dm}(\theta_h)$, GP based uncertainty $\mathcal{L}_{gp}(\theta_h, \theta_{gp})$ and RL objective $\mathcal{L}_{dpg}(\theta_h)$:

$$\pi_h = \underset{\theta_h}{\operatorname{argmin}} \mathcal{L}_d(\theta_h) := \mathcal{L}_{dm}(\theta_h) + \psi\mathcal{L}_{gp}(\theta_h, \theta_{gp}) + \eta\mathcal{L}_{dpg}(\theta_h), \tag{5}$$

where $\psi$ and $\eta$ are hyperparameters.

We adopt the objective proposed by Ho et al. (2020) as the diffusion objective,

$$\mathcal{L}_{dm}^h(\theta_h) = \mathbb{E}_{i \sim \mathcal{U}, \epsilon \sim \mathcal{N}(\boldsymbol{0}, \boldsymbol{I}), (\boldsymbol{s}, \boldsymbol{g}) \sim \mathcal{D}_h}\left[\left\|\boldsymbol{\epsilon} - \boldsymbol{\epsilon}_{\theta_h}\left(\sqrt{\bar{\alpha}_i}\boldsymbol{g} + \sqrt{1-\bar{\alpha}_i}\boldsymbol{\epsilon}, \boldsymbol{s}, i\right)\right\|^2\right], \tag{6}$$

where $\mathcal{D}_h$ is the high-level replay buffer, with the subgoals relabeled similarly to HIRO. Specifically, relabeling $\boldsymbol{g}_t$ in the high-level transition $(\boldsymbol{s}_t, \boldsymbol{g}_t, \sum_{i=t}^{t+k-1} R_i, \boldsymbol{s}_{t+k})$ with $\tilde{\boldsymbol{g}}_t$ aims to maximize the probability of the incurred low-level action sequence $\pi_l(\boldsymbol{a}_{t:t+k-1}|\boldsymbol{s}_{t:t+k-1}, \tilde{\boldsymbol{g}}_{t:t+k-1})$, which is approximated by maximizing the log probability

$$\log \pi_l(\boldsymbol{a}_{t:t+k-1}|\boldsymbol{s}_{t:t+k-1}, \tilde{\boldsymbol{g}}_{t:t+k-1}) \propto -\frac{1}{2}\sum_{i=t}^{t+k-1}||\boldsymbol{a}_i - \pi_{\theta_l}^l(\boldsymbol{s}_i, \tilde{\boldsymbol{g}}_i)||_2^2 + \text{const.} \tag{7}$$

The purpose of the above diffusion objective is to align the high-level policy's behavior with the distribution of "optimal" relabeled subgoals, thereby mitigating non-stationarity in hierarchical models.

While our method is versatile enough to be integrated with various actor-critic based HRL frameworks, we specifically utilize the TD3 algorithm (Fujimoto et al., 2018) at each hierarchical level, in line with precedents set by HIRO (Nachum et al., 2018) and HRAC (Zhang et al., 2020). Accordingly, the primary goal of the subgoal generator within this structured approach is to optimize for the maximum expected return as delineated by a deterministic policy. This objective is mathematically represented as:

$$\mathcal{L}_{dpg}(\theta_h) = -\mathbb{E}_{\mathbf{s} \sim \mathcal{D}_h, \mathbf{g}^0 \sim \pi_{\theta_h}}\left[Q_h(\mathbf{s}, \mathbf{g}^0)\right]. \tag{8}$$

Given that $\mathbf{g}^0$ (hereafter referred to as $\mathbf{g}$ without loss of generality) is reparameterized according to Eq. 4, the gradient of $\mathcal{L}_{dpg}$ with respect to the subgoal can be backpropagated through the entire diffusion process.

## 3.2 UNCERTAINTY MODELING

Whilst diffusion models possess sufficient expressivity to model the conditional distribution of subgoals, they face two significant challenges in the context of subgoal generation in HRL. Firstly, these models typically require a substantial amount of training data, *i.e.*, relabeled subgoals, to achieve accurate and stable performance, which can be highly sample inefficient. Secondly, standard diffusion models lack an inherent mechanism for quantifying uncertainty in their predictions, a crucial aspect for effective exploration and robust decision-making in HRL. To address these issues, we propose to model the state-conditional distribution of subgoals at the high level, harnessing a Gaussian Process (GP) prior as an explicit surrogate distribution for subgoals.

Given $\mathcal{D}_h$, the relabeled high-level replay buffer, a zero mean Gaussian process prior is placed on the underlying latent function, *i.e.*, the high-level policy, $\mathbf{g} \sim \pi_{\theta_h}$, to be modeled. This results in a multivariate Gaussian distribution on any finite subset of latent variables; in particular, at $\mathbf{X}$:

$$p(\mathbf{g}|\mathbf{s}; \theta_{gp}) = \mathcal{N}(\mathbf{g}|\mathbf{0}, \mathbf{K}_N + \sigma^2\mathbf{I}) \tag{9}$$

where the covariance matrix $\mathbf{V}$ is constructed from a covariance function, or kernel, $\mathbf{K}$ which expresses some prior notion of smoothness of the underlying function: $[\mathbf{K}_N]_{ij} = K(\mathbf{s}_i, \mathbf{s}_j)$. Typically, the covariance function depends on a small number of hyperparameters $\theta_{gp}$, which control these smoothness properties. Without loss of generality, we employ the commonly used Radial Basis Function (RBF) kernel:

$$K(\mathbf{s}_i, \mathbf{s}_j) = \gamma^2 \exp\left[-\frac{1}{2l^2}\sum_{d=1}^{D}\left(s_i^{(d)} - s_j^{(d)}\right)^2\right], \quad \theta_{gp} = \{\gamma, \ell, \sigma\}. \tag{10}$$

Here, $D$ is the state space dimension, $\gamma^2$ is the variance parameter, $\ell$ is the length scale parameter and $\sigma^2$ is the noise variance. $\theta_{gp}$ are learnable hyperparameters of the GP model.

We leverage the GP prior to regularize and guide the optimization of the diffusion policy. Specifically, we incorporate the negative log marginal likelihood of the GP as an additional loss term $\mathcal{L}_{gp}$:

$$\begin{aligned}\mathcal{L}_{gp} &= \mathbb{E}_{\mathbf{s}\sim\mathcal{D}_h, \mathbf{g}\sim\pi_{\theta_h}}\left[-\log p(\mathbf{g}|\mathbf{s}; \theta_h, \theta_{gp})\right] \\ &= \mathbb{E}_{\mathbf{s}\sim\mathcal{D}_h, \mathbf{g}\sim\pi_{\theta_h}}\left[-\frac{1}{2}\mathbf{g}^\top(\mathbf{K}_N + \sigma^2\mathbf{I})^{-1}\mathbf{g} - \frac{1}{2}\log|\mathbf{K}_N + \sigma^2\mathbf{I}| - \frac{N}{2}\log 2\pi\right],\end{aligned} \tag{11}$$

where $\mathbf{K}_N$ is the kernel matrix computed using the states $\mathbf{s}$. By minimizing $\mathcal{L}_{gp}$ alongside the diffusion loss and RL objective, we encourage the diffusion policy to generate subgoals that are consistent with the GP prior. This approach not only regularizes the diffusion model but also incorporates the uncertainty estimates provided by the GP, potentially leading to more robust and sample-efficient learning.

## 3.3 INDUCING STATES INFORMED EXPLORATION STRATEGY

Assuming the hyperparameters $\theta_{gp}$ are learned and fixed, the GP model predicting new subgoals is essentially determined by the state and subgoal pairs in $\mathcal{D}_h$. However, the training of GP requires $\mathcal{O}(N^3)$ time due to the inversion of the covariance matrix while the prediction is $\mathcal{O}(N)$ for the predictive mean and $\mathcal{O}(N^2)$ for the predictive variance per new state.

We now consider a pseudo dataset, $\bar{\mathcal{D}}$ (of size $M < N$) containing inducing states $\bar{\mathbf{s}}$ which parameterize the GP predictive distribution. We denote the "imaginary" subgoals $\bar{\mathbf{g}}$ and they are not real observations without including a noise variance for them. The single data point likelihood can thus be formulated as:

$$p(\mathbf{g}|\mathbf{s}, \bar{\mathcal{D}}, \bar{\mathbf{g}}) = \mathcal{N}\left(\mathbf{g}|\mathbf{k}_\mathbf{s}^\top\mathbf{K}_M^{-1}\bar{\mathbf{g}}, K_{\mathbf{ss}} - \mathbf{k}_\mathbf{s}^\top\mathbf{K}_M^{-1}\mathbf{k}_\mathbf{s} + \sigma^2\right), \tag{12}$$

where $[\mathbf{K}_M]_{mm'} = K(\bar{\mathbf{s}}_m, \bar{\mathbf{s}}_{m'})$ and $[\mathbf{k}_\mathbf{s}]_m = K(\bar{\mathbf{s}}_m, \mathbf{s})$, for $m = 1, \ldots, M$.
The subgoals are generated i.i.d. given the states, giving the overall likelihood of subgoals:

$$p(\mathbf{g}|\mathbf{s}, \bar{\mathbf{s}}, \bar{\mathbf{g}}) = \prod_{n=1}^{N} p(\mathbf{g}_n|\mathbf{s}_n, \bar{\mathbf{s}}, \bar{\mathbf{g}}) = \mathcal{N}(\mathbf{g}|\mathbf{K}_{NM}\mathbf{K}_M^{-1}\bar{\mathbf{g}}, \Lambda + \sigma^2\mathbf{I}), \tag{13}$$

where $[\Lambda]_{nn'} = K(\mathbf{s}_n, \mathbf{s}_{n'}) - \mathbf{k}_n^\top\mathbf{K}_M^{-1}\mathbf{k}_{n'}$ and $[\mathbf{K}_{NM}]_{nm} = K(\mathbf{s}_n, \bar{\mathbf{s}}_m)$.

|       | Reacher | Pusher | Point Maze | Ant Maze (U-shape) | Ant Maze (W-shape) | Stochastic Ant Maze (U) |
|-------|---------|--------|------------|--------------------|--------------------|-------------------------|
| HIDI  | **0.95±0.01** | **0.78±0.02** | **1.00±0.00** | **0.91±0.03** | **0.87±0.02** | **0.91±0.02** |
| HLPS  | 0.54±0.14 | 0.62±0.12 | 0.99±0.00 | 0.83±0.02 | 0.83±0.02 | 0.80±0.05 |
| SAGA  | 0.80±0.05 | 0.20±0.12 | 0.99±0.00 | 0.82±0.02 | 0.70±0.03 | 0.81±0.01 |
| HIGL  | 0.89±0.01 | 0.31±0.10 | 0.71±0.20 | 0.52±0.05 | 0.70±0.03 | 0.75±0.03 |
| HRAC  | 0.74±0.10 | 0.17±0.06 | 0.99±0.00 | 0.80±0.03 | 0.51±0.17 | 0.73±0.02 |
| HIRO  | 0.76±0.02 | 0.19±0.06 | 0.89±0.10 | 0.72±0.03 | 0.59±0.03 | 0.61±0.03 |

Table 1: Final performance of the policy obtained after 5M steps of training with sparse rewards, averaged over 10 randomly seeded trials with standard error.

Similar with Eq. (9), we assume a Gaussian prior for the imaginary subgols since they are expected to be distributed similarly to the real subgoals:

$$p(\bar{\mathbf{g}}|\bar{\mathbf{s}}) = \mathcal{N}(\bar{\mathbf{g}}|0, \mathbf{K}_M). \tag{14}$$

By applying Bayes rule on Eq. (13) and (14), we can derive the posterior distribution of the imaginary subgoals:

$$p(\bar{\mathbf{g}}|\mathbf{D}_h, \bar{\mathbf{s}}) = \mathcal{N}(\bar{\mathbf{g}}|\mathbf{K}_M\mathbf{Q}_M^{-1}\mathbf{K}_{MN}(\Lambda + \sigma^2\mathbf{I})^{-1}\mathbf{g}, \mathbf{K}_M\mathbf{Q}_M^{-1}\mathbf{K}_M), \tag{15}$$

where $\mathbf{Q}_M = \mathbf{K}_M + \mathbf{K}_{MN}(\Lambda + \sigma^2\mathbf{I})^{-1}\mathbf{K}_{NM}$.

Given a new state $\mathbf{s}_*$, the predictive distribution is then obtained by integrating the likelihood (12) with the posterior (15) (Snelson & Ghahramani, 2005):

$$p(\mathbf{g}_*|\mathbf{s}_*, \mathbf{D}_h, \bar{\mathbf{s}}) = \int d\bar{\mathbf{g}}\, p(\mathbf{g}_*|\mathbf{s}_*, \bar{\mathbf{s}}, \bar{\mathbf{g}})p(\bar{\mathbf{g}}|\mathbf{D}_h, \bar{\mathbf{s}}) = \mathcal{N}(\mathbf{g}_*|\mu_*, \sigma_*^2), \tag{16}$$

where

$$\mu_* = \mathbf{k}_*\mathbf{Q}_M^{-1}\mathbf{K}_{MN}(\Lambda + \sigma^2\mathbf{I})^{-1}\mathbf{g} \tag{17}$$

$$\sigma_*^2 = K_{**} - \mathbf{k}_*^\top(\mathbf{K}_M^{-1} - \mathbf{Q}_M^{-1})\mathbf{k}_* + \sigma^2 \tag{18}$$

The computational cost during training is dominated by the calculation of $\mathbf{Q}_M$, which is $O(M^2N)$; the prediction can be done in $O(M)$ for the mean and $O(M^2)$ for the variance per inference.

We are left with the problem of finding the pseudo-input locations $\bar{\mathbf{s}}$ and hyperparameters $\theta_{gp}$. We can do this by computing the marginal likelihood from (13) and (14):

$$p(\mathbf{g}|\mathbf{s}, \bar{\mathbf{s}}, \theta_{gp}) = \int d\bar{\mathbf{g}}\, p(\mathbf{g}|\mathbf{s}, \bar{\mathbf{s}}, \bar{\mathbf{g}}, \theta_{gp})p(\bar{\mathbf{g}}|\bar{\mathbf{s}}) = \mathcal{N}(\mathbf{g}|0, \mathbf{K}_{NM}\mathbf{K}_M^{-1}\mathbf{K}_{MN} + \Lambda + \sigma^2\mathbf{I}). \tag{9}$$

The marginal likelihood can then be maximized with respect to all these parameters $\{\bar{\mathbf{s}}, \theta_{gp}\}$ by gradient ascent, which encourages the pseudo state to adaptively summarize the states in order to facilitate finer modeling in critical regions of the state space. As indicated in Eq. 16, the learned inducing states provides implicit subgoals, which are used to model the distribution of diffusional subgoals.

Building upon the above inducing states based uncertainty modeling, we propose an exploration strategy that leverages the above inducing state based predictive distribution to guide subgoal selection during high-level policy execution. This method aims to enhance exploration efficiency and improve overall performance in HRL by utilizing the GP's predictive mean as an estimated promising subgoal.

Our strategy operates as follows: at each high-level decision point, with probability $\varepsilon$, we select the subgoal as the GP's predictive mean $\boldsymbol{\mu}_*$ (Eq. (17)) at the current state $\mathbf{s}_*$; otherwise, we sample a subgoal $\mathbf{g}_*$ from the diffusion policy $\pi_{\theta_h}$. Formally the subgoal generation follows:

$$\mathbf{g}_* = \begin{cases} \boldsymbol{\mu}_*, & \text{with probability } \varepsilon, \\ \mathbf{g} \sim \pi_{\theta_h}(\mathbf{g}\,|\,\mathbf{s}_*), & \text{with probability } 1 - \varepsilon. \end{cases} \tag{19}$$

This approach effectively biases the agent's exploration towards promising subgoals based on the GP predictive distribution harnessing inducing states, leveraging the model's uncertainty estimates to guide exploration in a more targeted, informed approach.

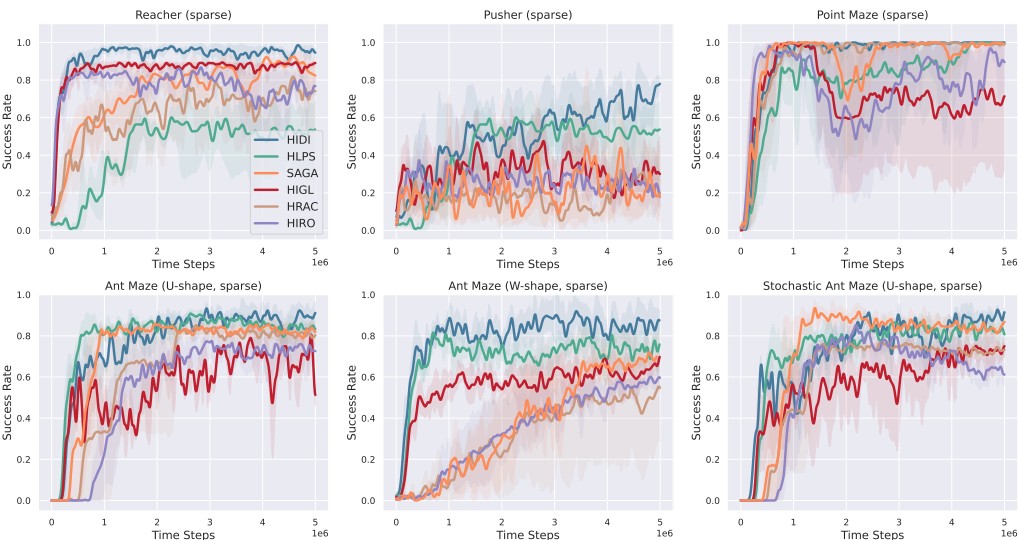

Figure 2: Learning curves of our method and baselines, *i.e.*, **HLPS** (Wang et al., 2024), **SAGA** (Wang et al., 2023), **HIGL** (Kim et al., 2021), **HRAC** (Zhang et al., 2020), and **HIRO** (Nachum et al., 2018), with sparse rewards. Each curve and its shaded region represent the average success rate and 95% confidence interval respectively, averaged over 10 independent trials.

# 4 RELATED WORK

HRL has been a significant field of study for addressing challenges such as long-term credit assignment and sparse rewards. It typically involves a high-level policy that breaks down the overarching task into manageable subtasks, which are then addressed by a more specialized low-level policy (Dayan & Hinton, 1992; Schmidhuber & Wahnsiedler, 1993; Kulkarni et al., 2016; Vezhnevets et al., 2017; Nachum et al., 2018; Levy et al., 2019; Zhang et al., 2020; Li et al., 2021; Kim et al., 2021; Li et al., 2023; Wang et al., 2023). The mechanism of this decomposition varies, with some approaches utilizing discrete values for option or skill selection (Bacon et al., 2017; Fox et al., 2017; Gregor et al., 2017; Konidaris & Barto, 2009; Eysenbach et al., 2019; Sharma et al., 2020; Bagaria & Konidaris, 2019), and others adopting learned subgoal space (Vezhnevets et al., 2017; Li et al., 2021; 2023). Despite these variations, a common challenge is the difficulty of leveraging advancements in the field of off-policy, model-free RL.

Recent efforts to enhance HRL's learning efficiency through off-policy training have highlighted issues such as instability and the inherent non-stationarity problem of HRL. For instance, Nachum et al. (2018) introduces an off-policy method that relabels past experiences to mitigate non-stationary effects on training. Techniques from hindsight experience replay have been utilized to train multi-level policies concurrently, penalizing high-level policies for unattainable subgoals (Andrychowicz et al., 2017a; Levy et al., 2019). To address the issue of large subgoal spaces, Zhang et al. (2020) proposed constraining the high-level action space with an adjacency requirement. Wang et al. (2020) improves stationarity by conditioning high-level decisions on both the low-level policy representation and environmental states. Li et al. (2021) develops a slowness objective for learning a subgoal representation function. Kim et al. (2021) introduces a framework for training a high-level policy with a reduced action space guided by landmarks, *i.e.*, promising states to explore. Adopting the deterministic subgoal representation of Li et al. (2021), Li et al. (2022) proposes an exploration strategy to enhance the high-level exploration via designing measures of novelty and potential for subgoals, albeit the strategy relies on on visit counts for subgoals in the constantly changing subgoal representation space.

The broader topic of goal generation in RL has also been explored (Florensa et al., 2018; Nair et al., 2018; Ren et al., 2019; Campero et al., 2021; Wang et al., 2023). GoalGAN (Florensa et al., 2018) employs a GAN to generate appropriately challenging tasks for policy training, however it does not condition on observations and the sequential training of its GAN and policy. Nair et al. (2018) combines unsupervised representation learning with goal-conditioned policy training. Ren et al. (2019) proposes a method for generating immediately achievable hindsight goals. Campero et al.

(2021) introduces a framework wherein a teacher network proposes progressively challenging goals, rewarding the network based on the student's performance. SAGA Wang et al. (2023) introduces an adversarially guided framework for generating subgoals in goal-conditioned HRL. However, akin to the common challenges associated with GANs, SAGA may encounter issues such as stability and mode collapse due to its implicit modeling of subgoal distributions. In contrast, HIDI explicitly constructs subgoal distributions by progressively transforming noise into sample data, effectively capturing multimodal distributions and ensuring more stable training dynamics.

Diffusion models have been introduced to offline RL domain recently (Janner et al., 2022; Wang et al.; Kang et al., 2024; Li et al., 2023; Chen et al., 2024). Janner et al. (2022) train an unconditional diffusion model to generate trajectories consisting of states and actions for offline RL. Approaches (Li et al., 2023; Chen et al., 2024) also extend diffusion model to offline HRL and generate trajectories at different levels. As a more related line of work, Diffsuion-QL (Wang et al.) introduces diffusion models into offline RL and demonstrated that diffusion models are superior at modeling complex action distributions. Kang et al. (2024) improve Diffsuion-QL to be compatible with maximum likelihood-based RL algorithms. The success of offline RL methods leveraging diffusion policies (Wang et al.; Kang et al., 2024) motivates us to investigate the impact of using conditional diffusion model for the challenging subgoal generation in off-policy HRL.

Gaussian processes can encode flexible priors over functions, which are a probabilistic machine learning paradigm (Williams & Rasmussen, 2006). GPs have been adopted in various latent variable modeling tasks in RL. In Engel et al. (2003), the use of GPs for solving the RL problem of value estimation is introduced. Then Kuss & Rasmussen (2003) uses GPs to model the the value function and system dynamics. Deisenroth et al. (2013) develops a GP-based transition model of a model-based learning system, which explicitly incorporates model uncertainty into long-term planning and controller learning to reduce the effects of model errors. Levine et al. (2011) proposes an algorithm for inverse reinforcement learning that represents nonlinear reward functions with GPs, allowing the recovery of both a reward function and the hyperparameters of a kernel function that describes the structure of the reward. Wang et al. (2024) proposes a GP based method for learning probabilistic subgoal representations in HRL.

## 4.1 ENVIRONMENTS

Our experimental evaluation encompasses a diverse set of long-horizon continuous control tasks facilitated by the MuJoCo simulator (Todorov et al., 2012), which are widely adopted in the HRL community. The environments selected for testing our framework, depicted in Figure 2, include:

- **Reacher:** This task entails utilizing a robotic arm to reach a specified target position with its end-effector.
- **Pusher:** A robotic arm must push a puck-shaped object on a plane to a designated goal position.
- **Point Maze:** A simulation ball starts in the bottom left corner of a " ⊃"-shaped maze, aiming to reach the top left corner.
- **Ant Maze (U-shape):** A simulated ant starts in the bottom left of a " ⊃"-shaped maze, targeting the top left corner.
- **Ant Maze (W-shape):** A simulated ant starts at a random position within a "∃"-shaped maze, aiming for the middle left corner.
- **Stochastic Ant Maze (U-shape):** Gaussian noise (standard deviation $\sigma = 0.05$) is added to the ant's $(x, y)$ position at each step.

We evaluate HIDI and all the baselines under two reward shaping paradigms: dense and sparse. In the dense setting, rewards are computed as the negative L2 distance from the current state to the target position within the goal space, whilst the sparse rewards are set to 0 for distances to the target below a certain threshold, otherwise -1. Maze tasks adopt a 2-dimensional goal space for the agent's $(x, y)$ position, adhering to existing works (Zhang et al., 2020; Kim et al., 2021). For Reacher, a 3-dimensional goal space is utilized to represent the end-effector's $(x, y, z)$ position, while Pusher employs a 6-dimensional space, including the 3D position of the object. Relative subgoal scheme is applied in Maze tasks, with absolute scheme used for Reacher and Pusher[1].

---

[1]Further details, including environment specifics, source code, and parameter settings for experiment reproduction, are provided in the appendix.

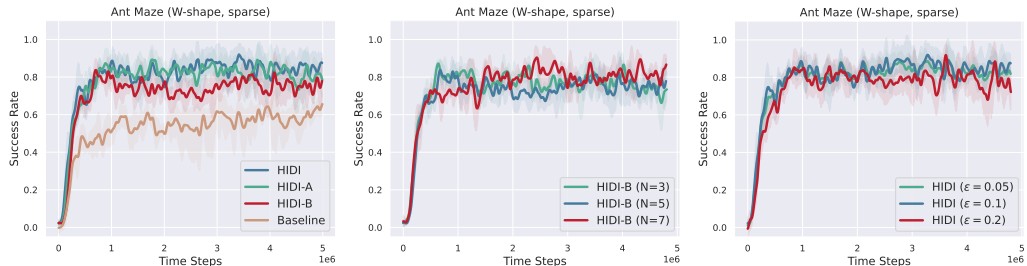

Figure 3: (Left) Learning curves of various baselines: HIDI-A refers to HIDI w/o exploration; HIDI-B refers to HIDI w/o exploration and GP priors. (Middle) HIDI with various diffusion steps. (Right) HIDI with various probability of performing explorations.

| | HIDI | SAGA | HIGL | HRAC | HIRO |
|---|---|---|---|---|---|
| $\Delta$ | **0.82±0.08** | 0.95±0.13 | 1.57±0.05 | 1.72±0.06 | 10.90± 2.04 |

Table 2: The distance between generated subgoals and the reached subgoals, *i.e.*, the final state of k-step low-level roll-out, averaged over 10 randomly seeded trials with standard error.

## 4.2 ANALYSIS

We conduct experiments comparing with the following state-of-the-art baseline methods: (1) **HLPS** (Wang et al., 2024): an HRL algorithm which proposes a GP-based subgoal latent space to address the non-stationarity issue; (2) **HESS** (Li et al., 2022): an HRL algorithm which proposes an exploration strategy to avoid introducing additional non-stationarity; (3) **SAGA** (Wang et al., 2023): an HRL algorithm that introduces an adversarially guided framework for generating subgoals; (4) **HIGL** (Kim et al., 2021): an HRL algorithm that trains a high-level policy with a reduced subgoal space guided by landmarks; (5) **HRAC** (Zhang et al., 2020): an HRL algorithm which introduces an adjacency network to restrict the high-level action space to a $k$-step adjacent region of the current state; (6) **HIRO** (Nachum et al., 2018): an HRL algorithm that relabels the high-level actions based on hindsight experience (Andrychowicz et al., 2017b). Additionally, we present a theoretical analysis for HIDI in the Appendix A.1.

**Can HIDI surpass state-of-the-art HRL methods in learning stability, sample efficiency, and asymptotic performance?** Figure 2 illustrates the learning curves of HIDI in comparison with baseline methods across various tasks. Details on Maze tasks with dense external rewards are included in the Appendix. HIDI consistently surpasses all baselines in terms of learning stability, sample efficiency, and asymptotic performance. The advantage of the hierarchical diffusion policy approach is more pronounced in complex scenarios such as the *Reacher* and *Pusher* robotic arm tasks, and the *Stochastic Ant Maze* task, where the environmental stochasticity poses additional challenges in generating reasonable subgoals. These results highlight the benefits of employing generative models for subgoal generation, as demonstrated by HIDI and SAGA. However, SAGA exhibits signs of instability in the more demanding robotic arm task Pusher, as well as considerably lower sample efficiency compared to HIDI in most tasks.

**Is HIDI capable of generating reachable subgoals to address the non-stationarity issue commonly encountered in off-policy training within HRL?** Figure 6 in the Appendix illustrates generated subgoals and reached subgoals of HIDI and compared baselines in Ant Maze (W-shape, sparse) with the same starting location. This allows for an intuitive comparison of subgoals generated by HIDI, SAGA, HIGL, HRAC, and HIRO. Notably, HIDI generates reasonable subgoals for the lower level to achieve, as demonstrated by the small divergence between the generated and reached subgoals, which provides a stable learning signal for the low-level policy. In contrast, subgoals generated by HIRO are often unachievable and fail to guide the agent to reach the final target; subgoals generated by HRAC frequently get stuck in local minima due to its local adjacency constraint; HIGL and SAGA show improvement in both constraining the subgoals locally while jumping out of the local optimum of subgoals, yet the high-level policy is not adequately compatible with the low-level skills, *i.e.*, the increasing gap between the generated subgoal and reached subgoal leads to inferior performance compared with HIDI. This is further confirmed by the measure of distance between the

generated subgoal and reached subgoal in Table 2. Subgoals generated by HLPS and HESS lie in a learned subgoal latent space and may not be quantitatively or qualitatively compared with other methods.

**How do various design choices within HIDI impact its empirical performance and effectiveness?**
To understand the benefits of using a diffusion policy for generating subgoals, we constructed several baselines. *HIDI-A* denotes a baseline without performing the proposed exploration strategy, *HIDI-B* is a baseline without adopting the GP regularization and exploration, while *Baseline* is HIDI without difusion model, GP regularization and exploration strategy. Fig. 3 (Left) illustrates the comparisons of various baselines and HIDI:

- Diffusional Subgoals: The performance improvement from *Baseline* to HIDI-B shows the benefit of adopting conditional diffusional model for subgoal generation, with a performance gain of $\sim15\%$.

- Uncertainty Regularization: The sampling efficiency and performance improvement $\sim4\%$ from HIDI-B to HIDI-A indicates the advantage of employing the GP prior on subgoal generation as a surrogate distribution which potentially informs the diffusion process about uncertain areas.

- Exploration Strategy: The comparison between HIDI and HIDI-A shows the benefit $\sim5\%$ of the proposed exploration strategy that identifies promising reachable subgoals.

- Diffusion Steps $N$: Fig. 3 (Middle) shows the learning curves of baselines using varying number of diffusion steps $N$ used in HIDI. It empirically demonstrates that as $N$ increases from 3 to 7, the high-level policy becomes more expressive and capable of learning the more complex data distribution of subgoals. Since $N$ also serves as a trade-off between the expressiveness of subgoal modeling and computational complexity, we found that $N = 5$ is an efficient and effective setting for all the tasks during the experiment.

- Exploration Probability $\epsilon$: $\epsilon$ controls the chance to perform explorations. As shown in Fig. 3 (Right), when $\epsilon$ is large, *e.g.*, 0.25, the trade-off between the expressiveness of subgoal modeling and uncertainty information might be affected, *i.e.*, excessive subgoals sampled in uncertain regions may contribute to performance instability. When $\epsilon$ is small, *e.g.*, 0.05, the gain from explorations would be decreased, and we set $\epsilon = 0.1$ for all other results.

- Scaling Factor $\eta$: We investigate the impact of $\eta$ in Eq. (5), which balances the diffusion objective and RL objective. As shown in Fig. 4 (Left), increasing $\eta$ improves performance at early training steps $(0 \sim 10^6)$, while all three settings achieve similar performance at larger training steps $(4.2 \times 10^6 \sim 5 \times 10^6)$. We report all other results based on $\eta = 5$ without loss of generality.

- Scaling Factor $\psi$: $\psi$ adjusts the influence of GP prior in learning the distribution of diffusional subgoals. As shown in Fig. 4 (Middle) in the appendix, increasing $\psi$ gives stronger GP prior and may slightly affect the flexibility of diffusion model learning, while decreasing $\psi$ renders the model to approximate baseline HIDI-B. We set $\psi = 10^{-3}$ for all other results.

**Is the proposed exploration strategy advantageous compared to the visit count-based exploration in HRL?** To investigate the advantage of the proposed exploration strategy, which adopts the predictive distribution of diffusional subgoals underpinned by learned inducing states, we compare HIDI with HESS, which also actively performs exploration at the high level based on the novelty, *i.e.*, the visit count, and potential, *i.e.*, negative distance between a sampled subgoal and the ending state of a low-level trajectory. As shown in Fig. 4, HIDI significantly outperforms HESS in the robotic arm tasks Reacher and Pusher, demonstrating higher sample efficiency. Our insight is that the exploration of HIDI, which is based on a non-parametric distribution, might be more data-efficient and stable than relying on visit counts for subgoals as in HESS.

## 5 Conclusion

We presented a hierarchical reinforcement learning framework that employs conditional diffusion models for subgoal generation to address non-stationarity issues in off-policy HRL training. By incorporating a Gaussian Process prior into the diffusion policy, we provided a probabilistic approach that quantifies uncertainty in the subgoal space, enhancing sample efficiency and exploration. Introducing adaptive inducing states allowed the model to focus on critical regions of the state space, improving learning performance. Our exploration strategy, based on the predictive distribution of diffusional subgoals, demonstrated significant advantages over existing methods, yielding superior results on challenging continuous control benchmarks.

**Reproducibility Statement** To ensure the reproducibility of our work, we have taken several measures throughout this paper. In the Implementation section, we provide detailed information about our two-layer hierarchical policy network architecture, including the use of TD3 as the foundational RL algorithm and the specific parameterization of the diffusion policies. We describe the MLP-based conditional diffusion model structure, including the number of layers, hidden units, and input composition. Further implementation details are provided in Table 3. The Algorithm section presents Algorithm 1, which outlines the complete training procedure for HIDI. To facilitate direct replication of our experiments, we have made our source code available [2]. This repository contains all necessary components to reproduce our results, including model architectures and hyperparameters.

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

## A  APPENDIX

### A.1  THEORETICAL ANALYSIS OF HIDI

We present a convergence analysis for the Hierarchical Diffusion-based Imitation (HIDI) algorithm in this section. We aim to establish that under certain conditions, HIDI converges to a policy that balances optimality and diversity in subgoal generation within a continuous hierarchical Markov Decision Process (MDP).

### A.1.1 PRELIMINARIES AND ASSUMPTIONS

Consider a continuous hierarchical MDP defined by $(\mathcal{S}, \mathcal{G}, \mathcal{A}, P, R, \gamma)$, where:

- $\mathcal{S} \subset \mathbb{R}^d$ is the state space.
- $\mathcal{G} \subset \mathbb{R}^m$ is the subgoal space.
- $\mathcal{A}$ is the action space.
- $P : \mathcal{S} \times \mathcal{G} \times \mathcal{S} \to [0, 1]$ is the transition probability density function.
- $R : \mathcal{S} \times \mathcal{G} \to \mathbb{R}$ is the reward function.
- $\gamma \in (0, 1)$ is the discount factor.

We introduce the following assumptions:

[Assumption 1: Compact Spaces] The state space $\mathcal{S} \subset \mathbb{R}^d$ and the subgoal space $\mathcal{G} \subset \mathbb{R}^m$ are compact and convex.

[Assumption 2: Bounded Rewards] The reward function $R : \mathcal{S} \times \mathcal{G} \to \mathbb{R}$ is bounded. That is, there exists $R_{\max} > 0$ such that for all $\mathbf{s} \in \mathcal{S}$ and $\mathbf{g} \in \mathcal{G}$,

$$|R(\mathbf{s}, \mathbf{g})| \leq R_{\max}. \tag{20}$$

[Assumption 3: Lipschitz Continuity] The reward function $R(\mathbf{s}, \mathbf{g})$, the transition probability density $P(\mathbf{s}'|\mathbf{s}, \mathbf{g})$, and the policy $\pi_{\theta_h}(\mathbf{g}|\mathbf{s})$ are Lipschitz continuous in their arguments. Specifically, there exist constants $L_R$, $L_P$, and $L_\pi$ such that:

$$|R(\mathbf{s}, \mathbf{g}) - R(\mathbf{s}', \mathbf{g}')| \leq L_R \left( \|\mathbf{s} - \mathbf{s}'\| + \|\mathbf{g} - \mathbf{g}'\| \right), \tag{21}$$

$$|P(\mathbf{s}''|\mathbf{s}, \mathbf{g}) - P(\mathbf{s}''|\mathbf{s}', \mathbf{g}')| \leq L_P \left( \|\mathbf{s} - \mathbf{s}'\| + \|\mathbf{g} - \mathbf{g}'\| \right), \tag{22}$$

$$|\pi_{\theta_h}(\mathbf{g}|\mathbf{s}) - \pi_{\theta_h}(\mathbf{g}'|\mathbf{s}')| \leq L_\pi \left( \|\mathbf{s} - \mathbf{s}'\| + \|\mathbf{g} - \mathbf{g}'\| \right). \tag{23}$$

[Assumption 4: Learning Rates] The sequences of learning rates $\{\alpha_k\}_{k=1}^{\infty}$ and $\{\beta_k\}_{k=1}^{\infty}$ for the critics and the policy, respectively, satisfy:

$$\sum_{k=1}^{\infty} \alpha_k = \infty, \quad \sum_{k=1}^{\infty} \alpha_k^2 < \infty; \tag{24}$$

$$\sum_{k=1}^{\infty} \beta_k = \infty, \quad \sum_{k=1}^{\infty} \beta_k^2 < \infty. \tag{25}$$

[Assumption 5: Function Approximation] The function approximators used for the critics and the policy (e.g., neural networks) are sufficiently expressive to approximate any continuous function on $\mathcal{S} \times \mathcal{G}$ arbitrarily closely.

### A.1.2 KEY COMPONENTS AND LEMMAS

**Diffusion Model Policy**    The high-level policy $\pi_{\theta_h}(\mathbf{g}|\mathbf{s})$ is modeled using a conditional diffusion model (Sohl-Dickstein et al., 2015; Ho et al., 2020):

$$\pi_{\theta_h}(\mathbf{g}|\mathbf{s}) = p_{\theta_h}(\mathbf{g}^{0:N}|\mathbf{s}) = p(\mathbf{g}^N) \prod_{i=1}^{N} p_{\theta_h}(\mathbf{g}^{i-1}|\mathbf{g}^i, \mathbf{s}). \tag{26}$$

Here, $p(\mathbf{g}^N) = \mathcal{N}(\mathbf{g}^N; \mathbf{0}, \mathbf{I})$ is the prior distribution.

**Lemma 1** (Convergence of the Diffusion Model Policy). *Under Assumptions (1, 3, 5), and given sufficient training data and model capacity, the diffusion model policy $\pi_{\theta_h}(\mathbf{g}|\mathbf{s})$ converges to an approximation of the true conditional subgoal distribution, balancing optimality and diversity.*

*Proof.* This follows from the consistency of maximum likelihood estimation in deep generative models (Song et al., 2021b). Given sufficient data and model capacity, and assuming that the model is trained to convergence using stochastic gradient descent with appropriate learning rates, the learned distribution $\pi_{\theta_h}(\mathbf{g}|\mathbf{s})$ approximates the true data distribution conditioned on $\mathbf{s}$ (Kingma & Welling, 2019). □

**TD3 Regularization**   The diffusion model is regularized using a Twin Delayed Deep Deterministic Policy Gradient (TD3) objective (Fujimoto et al., 2018).

**Lemma 2** (Approximate Convergence of Twin Critics). *Under Assumptions (1-5), and assuming that the function approximators are trained using stochastic gradient descent with diminishing learning rates (Assumption (4)), the twin critics $Q_{\phi_1}$ and $Q_{\phi_2}$ converge to functions that approximate the fixed point $Q^*$ satisfying the Bellman equation within a bounded error.*

*Proof.* Building upon the convergence analysis of temporal-difference learning with function approximation (Tsitsiklis & Roy, 1997), and considering the modifications in TD3 to mitigate overestimation bias (Fujimoto et al., 2018), we argue that under appropriate conditions, the critics minimize the mean squared Bellman error. The diminishing learning rates ensure that the error does not accumulate, leading to convergence to functions close to $Q^*$. □

**GP-based Uncertainty Modeling**   A Gaussian Process (GP) prior with learned inducing points is used to model the uncertainty in the subgoal distribution (Snelson & Ghahramani, 2006).

**Lemma 3** (Convergence of the GP Model). *Under Assumptions (1) and (5), and given sufficient data, the sparse GP model converges to an accurate approximation of the true subgoal uncertainty distribution as the number of training samples and inducing points increases.*

*Proof.* Based on the properties of sparse GP approximations (Titsias, 2009), as the number of inducing points and data samples increases, and assuming the inducing points are optimized effectively, the sparse GP model's predictive distribution converges to that of a full GP, accurately capturing the uncertainty. □

**Exploration Strategy**   The exploration strategy combines $\varepsilon$-greedy selection from the GP predictive mean and sampling from the diffusion model to ensure sufficient exploration.

**Lemma 4** (Sufficient Exploration). *Assuming that the exploration policy injects sufficient stochasticity into the action selection process, for any $\delta > 0$, every $\delta$-ball in $\mathcal{S} \times \mathcal{G}$ is visited infinitely often with probability one.*

*Proof.* By ensuring a non-zero probability of selecting any subgoal in $\mathcal{G}$ from any state $\mathbf{s}$ due to the stochastic component of the exploration strategy, the induced Markov chain is irreducible and aperiodic. According to the ergodicity property of Markov chains (Meyn & Tweedie, 1993), every state-subgoal pair is visited infinitely often. □

### A.1.3   MAIN CONVERGENCE THEOREM

**Theorem 5** (Convergence of HIDI). *Under Assumptions (1-5), and the conditions specified in Lemmas 1–4, the HIDI algorithm converges with high probability to a policy $\pi_{\theta_h}^*$ that balances optimality and diversity in subgoal generation, achieving performance within a bounded error of the optimal expected cumulative reward in the continuous hierarchical MDP.*

*Proof.* We establish convergence through the following steps:

**1) Sufficient Exploration**: Lemma 4 ensures that the algorithm explores the state-subgoal space thoroughly, preventing premature convergence to suboptimal policies due to limited exploration.

**2) Value Estimation Error Bounds**: From Lemma 2, the critics converge to approximations of the true value function within bounds determined by the function approximation capacity and learning dynamics.

**3) Policy Improvement with Approximation Errors**: Due to the approximation capabilities of the diffusion model and GP (Lemmas 1 and 3), the policy improvement step optimizes the expected return considering both the estimated values and uncertainties, within certain error margins.

**4) Error Propagation Control**: By carefully controlling the learning rates (Assumption (4)) and utilizing the stability provided by Lipschitz continuity (Assumption (3)), we ensure that errors introduced at each step do not accumulate excessively.

**5) Overall Convergence**: Combining the above, HIDI converges to a policy whose performance is close to the optimal policy, with the discrepancy bounded by the cumulative approximation errors from its components.

Formally, let $J(\pi) = \mathbb{E}\left[\sum_{t=0}^{\infty} \gamma^t R(\mathbf{s}_t, \mathbf{g}_t)\right]$ denote the expected return under policy $\pi$. Then, the policy $\pi^*_{\theta_h}$ learned by HIDI satisfies:

$$|J(\pi^*_{\theta_h}) - J(\pi^\dagger)| \leq \epsilon, \tag{27}$$

where $\pi^\dagger$ is the optimal policy, and $\epsilon$ is a bound that depends on the approximation errors from the diffusion model, critics, and GP model. $\square$

### A.1.4 DISCUSSION

This convergence result demonstrates that HIDI effectively combines the strengths of diffusion models, TD3, and GP-based uncertainty modeling to learn a high-level policy that generates diverse yet reward-maximizing subgoals in a hierarchical reinforcement learning setting. The key is balancing exploration and exploitation while managing approximation errors through careful design and adherence to the stated assumptions.

### A.2 IMPLEMENTATION

We implement the two-layer hierarchical policy network following the architecture of the HRAC Zhang et al. (2020), which uses TD3 Fujimoto et al. (2018) as the foundational RL algorithm for both the high and low levels.

Following the parameterization of Ho et al. (2020), the diffusion policies are implemented as an MLP-based conditional diffusion model, which is a residual model, *i.e.*, $\epsilon_\theta(g^i, s, i)$ and $\epsilon_\theta(a^i, s, g, i)$ respectively, where $i$ is the previous diffusion time step, $s$ is the state condition, and $g$ is the subgoal condition. $\epsilon_\theta$ is implemented as a 3-layer MLP with 256 hidden units. The inputs to $\epsilon_\theta$ comprise the concatenated elements of either the low-level or high-level action from the previous diffusion step, the current state, the sinusoidal positional embedding of time step $i$, and the current subgoal if it is a high-level policy. We provide further implementation details used for our experiments in Table 3.

### A.3 ALGORITHM

We provide Algorithm 1 to show the training procedure of HIDI. We provide the source code at https://anonymous.4open.science/r/HIDI-32F6/.

### A.4 ADDITIONAL RESULTS

Figure 6 shows the generated subgoals and reached subgoals of HIDI and compared baselines in Ant Maze (W-shape, sparse) with the same starting location. HIDI generates reasonable subgoals for the lower level to accomplish, as evidenced by the minimal divergence between the generated and achieved subgoals which in turn provides a stable learning signal for the low-level policy. In contrast, the subgoals generated by HIRO are unachievable and cannot guide the agent towards the final target. Subgoals from HRAC frequently get stuck in local minima due to its local adjacency constraint. While HIGL and SAGA demonstrate improvement in constraining subgoals locally while escaping local optima, the high-level policy is inadequately compatible with the low-level skills. Specifically, the increasing gap between the generated subgoal and reached subgoal leads to inferior performance compared to HIDI.

We also show the learning curves of our method and baselines in the aforementioned environments with dense rewards in Fig. 7, and its quantitative evaluation results can be found in Table 4.

We qualitatively study the learned inducing states in Fig. 5 (Right), where the states of the complete set of training batch (100) and the final learned inducing states (16) are visualized. Note, only the $x, y$ coordinates are selected from the state space in both cases for visualization purpose. We can observe that with a significantly smaller number of training data, the inducing points capture the gist of the complete training data by adapting to cover more critical regions of the state space, *e.g.*, the turning points in the Ant Maze (W-shape) task.

| Module | Parameter | Value |
|---|---|---|
| Diffusional Policy | Number of hidden layers | 1 |
| | Number of hidden units | 256 |
| | Nonlinearity, | Mish |
| | Optimizer | Adam |
| | Learning rate | $10^{-4}$ |
| | Hyperparameter for RL objective $\eta$ | 5 |
| | Number of diffusion steps $N$ | 5 |
| | GP loss weight | $10^{-3}$ |
| | GP learning rate | $3 \times 10^{-4}$ |
| | Exploration probability $\varepsilon$ | 0.1 |
| | Number of inducing states | 16 |
| Two-layer HRL, critic networks | Number of hidden layers | 1 |
| | Number of hidden units per layer | 300 |
| | Nonlinearity | ReLU |
| | Optimizer | Adam |
| | Learning rate, critic | $10^{-3}$ |
| | Batch size, high level | 100 |
| | Batch size, low level | 128 |
| | Replay buffer size | $2 \times 10^5$ |
| | Random time steps | $5 \times 5^6$ |
| | Subgoal frequency | 10 |
| | Reward scaling, high level | 0.1 |
| | Reward scaling, low level | 1.0 |

Table 3: Network architecture and key hyperparameters of HIDI

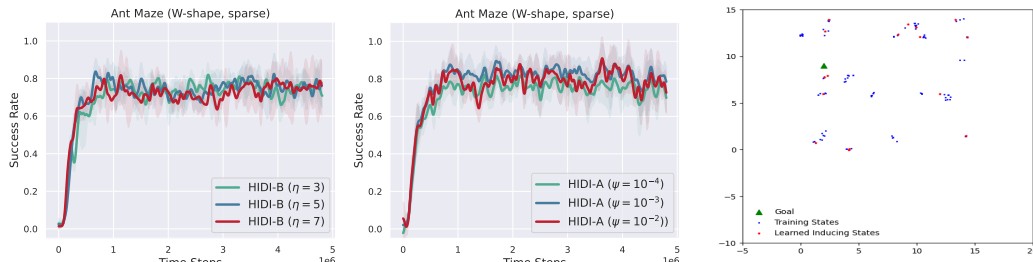

Figure 4: (Left) Impact of $\eta$, which balances the diffusion objective and RL objective. (Middle) Impact of $\psi$, which adjusts the influence of GP prior in learning the distribution of diffusional subgoals. (Right) Visualization of the learned inducing states (2D coordinates) compared with the complete training data.

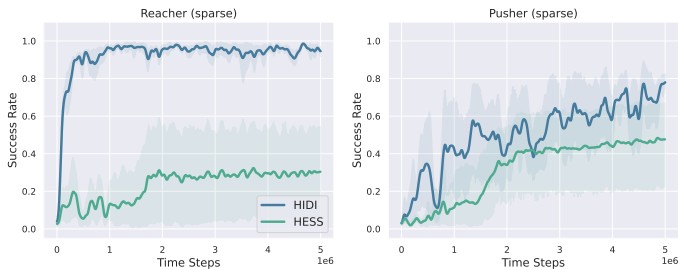

Figure 5: (Left) Learning curves of HIDI and HESS with an active exploration strategy.

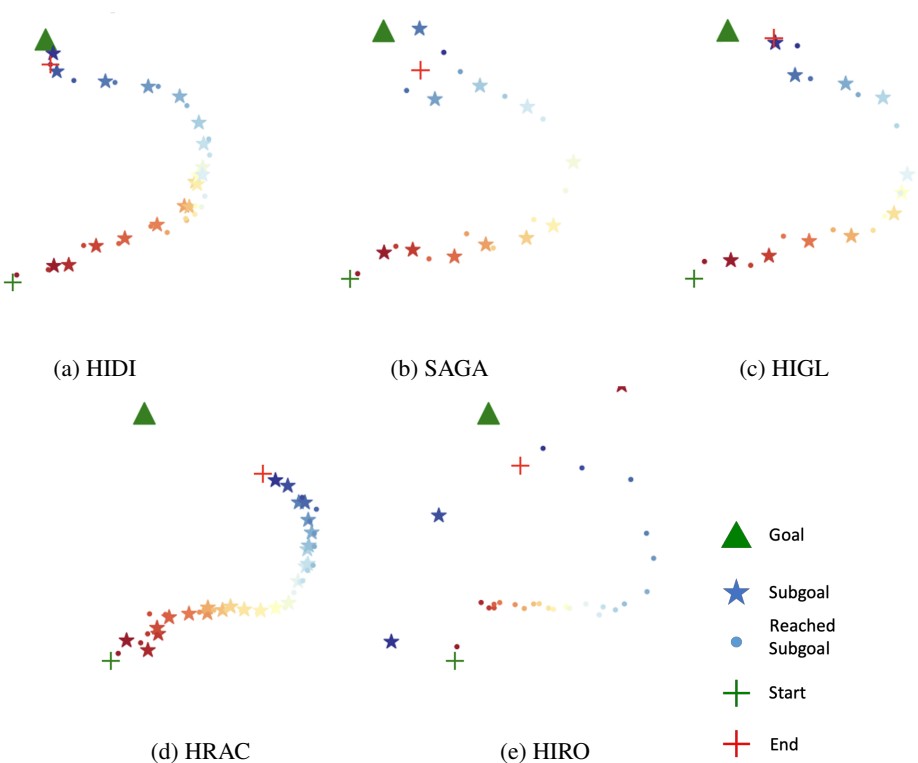

Figure 6: Visualization of generated subgoals and reached subgoals of HIDI and compared baselines in Ant Maze (W-shape, sparse) with the same starting location.

---

**Algorithm 1** HIDI

---

**Input:** Higher-level actor $\pi^h_{\theta_h}$, lower-level actor $\pi^l_{\theta_l}$, parameterized by $\theta_h$, $\theta_l$, respectively; critics $Q^h$ and $Q^l$; goal transition function $h(\cdot)$; higher-level action frequency $k$; number of training episodes $N$.

**for** $n = 1$ to $N$ do **do**
    Sample the initial state $s_0$ after resetting the environment.
    $t = 0$
    **repeat**
    **if** $t \equiv 0 \pmod{k}$ **then**
        With a probability of $\varepsilon$, explore with the strategy or sample subgoal $g_t \sim \pi^h_{\theta_h}(g|s_t)$, by Eq. (19)
    **else**
        Subgoal transition $g_t = h(g_{t-1}, s_{t-1}, s_t)$
    **end if**
    Sample lower-level action $a_t \sim \pi^l_{\theta_l}(a|s_t, g_t)$
    Execute $a_t$ and obtain next state $s_{t+1} \sim \mathcal{P}(s|s_t, a_t)$
    Obtain intrinsic reward $r_t \sim \mathcal{R}(r|s_t, g_t, a_t)$
    Store transition $(s_{t-1}, g_{t-1}, a_t, r_t, s_t, g_t)$ in replay buffer.
    Sample episode end signal *done*
    $t = t + 1$
    **until** *done* is *true*
    **if** Train higher-level policy $\pi^h_{\theta_h}$ **then**
        Sample experience $(s_t, \tilde{g}_t, \sum r_{t:t+k-1}, s_{t+k})$, where $\tilde{g}_t$ is relabeled subgoal by Eq. (7)
        Update high-level policy and GP hyperparameters with Eq. (5)
        Update higher-level critic $Q^h$ with experience
    **end if**
    **if** Train low-level policy $\pi^l_{\theta_l}$ **then**
        Sample experience $(s_t, a_t, g_t, r_t, s_{t+1})$
        Update low-level policy
        Update low-level critic $Q^l$ with experience
    **end if**
**end for**

---

| | Point Maze | Ant Maze (U-shape) | Ant Maze (W-shape) | Stochastic Ant Maze (U) |
|---|---|---|---|---|
| HIDI | **1.00±0.00** | **0.88±0.01** | **0.88±0.05** | **0.92±0.01** |
| HLPS | 1.00±0.00 | 0.83±0.01 | 0.80±0.02 | 0.88±0.03 |
| SAGA | 0.94±0.04 | 0.80±0.04 | 0.68±0.03 | 0.87±0.03 |
| HIGL | 0.98±0.02 | 0.83±0.07 | 0.78±0.04 | 0.82±0.03 |
| HRAC | 0.99±0.00 | 0.76±0.04 | 0.75±0.07 | 0.70±0.04 |
| HIRO | 0.81±0.19 | 0.75±0.07 | 0.50±0.04 | 0.80±0.03 |

Table 4: Final performance of the policy obtained after 5M steps of training with dense rewards, averaged over 10 randomly seeded trials with standard error. Comparisons are to **HLPS** (Wang et al., 2024), **SAGA** (Wang et al., 2023), **HIGL** (Kim et al., 2021), **HRAC** (Zhang et al., 2020), and **HIRO** (Nachum et al., 2018).

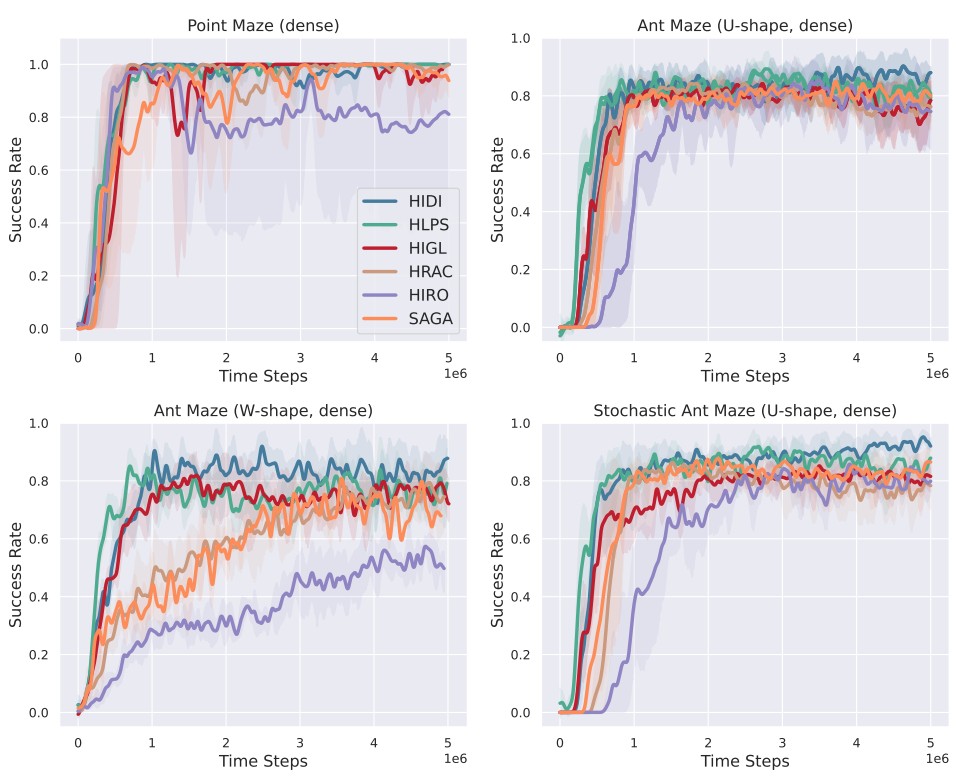

Figure 7: Learning curves of our method and baselines with dense rewards. Each curve and its shaded region represent the average success rate and 95% confidence interval respectively, averaged over 10 independent trials.