# OpenReview forum: "Uncertainty-Regularized Diffusional Subgoals for Hierarchical Reinforcement Learning"
_ICLR.cc/2025/Conference — ICLR 2025 Conference Withdrawn Submission_

### Official Review · Reviewer_frBb · 2024-10-25

**Soundness:** 3
**Presentation:** 2
**Contribution:** 3
**Rating:** 6
**Confidence:** 2

**Summary:**

This paper discusses the problem of subgoal generation in HRL. With a diffusion model as the high-level policy and a GP prior for subgoal distribution estimate, the proposed method efficiently outperforms baseline methods on various HRL benchmarks.

**Strengths:**

1. The problem of subgoal selection is fundamental in HRL. It is directly related to the sample efficiency and learning stability of HRL.
2. The idea of introducing diffusion models and GP is sensible. The authors assume that diffusion models are superior at fitting the subgoal distribution, and GP makes diffusion model learning easier. Empirical results support this calim.

**Weaknesses:**

1. HIDI's exploration is buit upon the idea of uncertainty. However, state space uncertainty struggles on high-deminsional tasks, such as MDPs with image observations. I would like to see more discussion on this issue.

**Questions:**

1. Can HIDI scale to MDPs with high-dimensional state spaces? E.g., the image-observation ant-maze mentioned in https://openreview.net/pdf?id=wxRwhSdORKG.

---

### Official Review · Reviewer_ZVuQ · 2024-11-03

**Soundness:** 3
**Presentation:** 2
**Contribution:** 2
**Rating:** 3
**Confidence:** 3

**Summary:**

This paper aims to address non-stationarity issues in off-policy training by introducing a conditional diffusion model for subgoal generation, which is regularized with a Gaussian Process (GP) prior. The GP helps quantify uncertainties in the action space, and adaptive inducing states are introduced to facilitate efficient subgoal generation. This approach aims to improve sample efficiency and exploration in hierarchical reinforcement learning. Experimental results show that the proposed method outperforms state-of-the-art HRL algorithms in continuous control benchmarks.

**Strengths:**

- The use of diffusion models and GP priors in hierarchical reinforcement learning for subgoal generation appears novel, combining generative modeling with uncertainty quantification to mitigate non-stationarity in HRL.
- The focus on addressing non-stationarity in off-policy HRL is an important challenge in the field. The approach, if well-validated, could have broader applications in enhancing the performance of hierarchical learning agents in complex environments.
- The method presented is extendable and can be easily combined with other reinforcement learning algorithms, making it a flexible solution for a variety of hierarchical learning problems.
- The ablation study is conducted in great detail, which helps identify the effectiveness of individual components.

**Weaknesses:**

- The paper does not well explain how diffusion models help to address the non-stationarity issue in HRL.
- The impact of exploration and the GP-based uncertainty modeling on performance is not adequately analyzed. Figure 3 shows HIDI without explroation has similar performance as HIDI.
- The proposed approach relies on Gaussian Processes, which typically face scalability issues as the number of training points increases. Although inducing points are introduced, there is no detailed discussion or analysis of how well the approach scales to larger.
- The environments used in the experiments are limited. This makes it difficult to evaluate the improvement of the proposed method compared to previous algorithms comprehensively.
- In environments with high randomness, such as the stochastic Ant Maze, the advantages of the proposed approach do not seem to be very apparent.

**Questions:**

- Can you explain how diffusion models help to address the non-stationarity issue?
- The subscript of the first $\epsilon$ in eq(4) does not seem to be displayed correctly. Should the last $\epsilon=0$ be $g^{i-1}=0$?
- What is the different between your diffusion model-based subgoal generator and diffusion policy[1]?
- Baseline in Fig. 3 seems to be weaker than algorithms in Fig. 2. Can your method be combined with algorithms in Fig. 2 like HLPS and SAGA?
- How good is HIDI when evaluating on other stochastic environments like Ant Fall (sparse reward) and Ant FourRooms (sparse reward)?
- How are the pseudo dataset $\overline D$ and the imaginary goal $\overline g$ obtained?
- In Fig. 3, HIDI and HIDI-A seem to have very small difference. What is the reason?
- Can you compare HIDI with other diffusion model-based algorithms listed in Sec. 4?
- In the main text, it is stated to compare HIDI with HESS, but no results about HESS are shown.

---

### Official Review · Reviewer_UfB6 · 2024-11-13

**Soundness:** 2
**Presentation:** 3
**Contribution:** 2
**Rating:** 3
**Confidence:** 3

**Summary:**

The paper tries to address a problem in HRL - the generation of subgoals while mitigating the non-stationarity issues in the hierarchies of policies. They propose using conditional diffusion models with Gaussian Process (GP) priors for the subgoal generation process. They claim diffusion process helps mitigate non-stationarity in hierarchical models, and GPs help with regularizing and measuring the uncertainty of the diffusion process, which makes the subgoal generation robust and sample-efficient. The experimental results show improvement over the previous methods.

**Strengths:**

- I liked the usage of diffusion models for the robustness, making it invariant to the noise.
- The use of GPs for the subgoal generations: a neat way to quantify the relationship between the states and goals to suggest new ones.
- I liked the usage of confidence intervals with a reasonable number of independent runs for the experimental results.
- Good ablation studies in the experiments section.
- I like the comparison of results of non-parametric distributions with count-based ones.

**Weaknesses:**

1. The authors state (line 204) that the diffusion objective aids in aligning the high-level policy’s behavior with the "optimal" relabeled subgoals, thus addressing non-stationarity. However, they provide no motivation, examples, or proof to clarify how or why the diffusion objective achieves this alignment.
2. The authors claim (line 249) that Gaussian Process (GP) priors provide uncertainty measures, regularize the diffusion process, and enhance the robustness and sample efficiency in learning. While the uncertainty aspect is clear, the other claims about regularization, robustness, and sample efficiency remain unconvincing.
3. Just as there is little clear motivation for using GP priors, the rationale for the proposed exploration strategy is also unclear—particularly why $\mu^*$ should be considered a good estimate of promising subgoals. Based on the results in Figure 3, HIDI and HIDI-A perform similarly, which suggests that including this exploration strategy may be redundant.
4. I don’t think that providing an external reward of -1 per step qualifies as sparse. It might appear sparse for the high-level policy since it achieves 0 only when it is near the goal, but it still provides useful feedback for subgoal learning via intrinsic reward. If the reward were instead set to 0 for each time step, that would make the problem genuinely sparse and challenging to solve.
5. The assertion that "increasing is better," as demonstrated in Figure 3 (middle) on lines 504-508, seems unconvincing since there are instances where  $N=3$ slightly outperforms  $N=5$.
6. The ablation studies were only conducted on Ant Maze environment, and I can’t find the other even in the appendix.

**Minor nitpicks**:

1. In Figure 3 (right) and Figure 4, all settings appear identical to me, so the statements in lines 509-520 seem redundant.
2. Consider using distinct letters for the different diffusion models. Having two instances of $\epsilon$ with different meanings is confusing.
3. The $N$ is not the right format on line 176.
4. The underscore for $\epsilon$  is small on line 185.

**Questions:**

1. How does the diffusion models help with the non-stationarity issue?
2. Why do you keep the covariance matrix fixed in the diffusion process? You assume the random variable are independent from each other, but I see no justification for that.
3. Why do you use reparameterization of the subgoals with diffusion models in the deterministic policy gradient objective?
4. Do you relabel the subgoals to ones that maximize the log probability as depcited on line 201?
5. Is the $N$ in the diffusion process different from one in the Gaussian Processes?
6. It is still unclear to me how you choose the $\bar{s}$. Do you randomly sample some representative points, or are they subset of the $\mathcal{D}_h$, or something else?
7. No mention of the sparse reward threshold. How is it set?

---

### Official Review · Reviewer_mCwz · 2024-11-13

**Soundness:** 1
**Presentation:** 2
**Contribution:** 2
**Rating:** 3
**Confidence:** 3

**Summary:**

The paper claims to address the well known hierarchical reinforcement learning(HRL) issue of non-stationarity when having a higher level policy learn off-policy from changing lower level policies. They do this by taking an existing HRL algorithm (HIRO) and using a diffusion model to generate subgoals, while adding a Gaussian Process to handle uncertainties of subgoals.

**Strengths:**

* The paper tackles a relevant problem in the field of HRL
* A good use of intrinsic reward to learn low-level policies.
* The authors compare their method with relevant baselines. They have a baseline that uses GP subgoal latent spaces, exploration techniques etc.
* The related work provides a concise but relevant survey of existing HRL, subgoal generation, and GP-based techniques in reinforcement learning (RL).
* The ablation study was useful at identifying which components of HIDI were responsible for performance increases.

**Weaknesses:**

1. The paper poses a problem of off-policy learning in the abstract and introduction, which was the same problem HIRO was designed to solve. However, the contributions of this paper upon HIRO does not appear to address the non-stationarity issue. Instead they propose a diffusion model as a subgoal generator, without motivating how such a model can address the off-policy problem.
2. Additionally, their second contribution, the GP prior seems to model the uncertainty over the subgoals. This was not directly related to the original off-policy problem, rather to help with uncertainty estimation on these diffusional subgoals.
3. The paper claims the GP model “potentially reduces the amount of training data needed for the diffusional model to converge”, but this was never tested. An experiment that could verify whether this is the case would be very useful for researchers seeking to incorporate diffusion as a generative model in reinforcement learning.
4. Line 250 is needlessly speculative: GPs were claimed “potentially leading to more robust and sample-efficient learning”, but Figure 3 shows that ablating the GP does not result in any significant performance drop.
5. It was stated that “diffusion models typically require a substantial amount of training data”, it would have been nice if the paper cited instances of this problem happening.
6. The paper claimed HIDI was more sample efficient than a count-based alternative (HESS), but they only show the performance of the two algorithms, no metric measuring exploration. Could HIDI be more sample efficient because it is exploring less? Since exploration is not directly related to the main problem the paper is addressing, this is not detrimental, but the reader may find the result distracting.
7. The paper’s last contribution of using inducing states is a clever trick to reduce the computational complexity of the GP, but they claim this gives better exploration. This was not empirically tested.
8. The exploration aspect of the sparse GP seems orthogonal to the paper’s initial objective of addressing the non-stationarity of a changing low level policy. Wouldn’t adding exploration make the higher policy training even more off-policy, as the low level policy has an epsilon chance of exploring?
9. The Environments and Empirical Analyses are written as subsections of the Related Works section. This should be a separate section.
10. It should be clarified that all the claims of “How do various design choices within HIDI impact its empirical performance and effectiveness?” are for the AntMaze W env

**Minor Fixes**
1. Is the sparse reward setting really sparse? The -1 per step signal is still informative for the lower level policy which reaches subgoals. A sparser reward could be 0 everywhere and +1 at the main goal.
2. Line 176: make the N calligraphic and the \mu and \Sigma boldface to match the equation.
3. Line 179: The learnt noise \boldsymbol{\epsilon}_{\theta_h} was introduced, but never defined anywhere. This may be standard notation with diffusion models, but having it explained (even in the appendix) can help readers with a reinforcement learning background understand.
4. Line 227: Introduces \mathbf{X}, but never defines or uses this matrix anywhere.
5. Line 254: Typo “Assuming the hyperparameters θgp are learned and fixed” -> “Assuming the hyperparameters θgp are learned and then fixed”
6. Line 259: This sentence needs to be re-structured, as its meaning is ambiguous. “We denote the “imaginary” subgoals g ̄ and they are not real observations without including a noise variance for them.”
7. Line 423: This paragraph gives details of the dimensionality of the goal spaces across envs, it may also help the reader to know what the dimension of the original state/observation space is for each environment.

**Questions:**

1. GPs were introduced because “standard diffusion models lack an inherent mechanism for quantifying uncertainty in their predictions”. Why couldn’t a learned covariance matrix, instead of the fixed \beta_i\mathbf{I}, be used instead of introducing a GP to model uncertainty?
2. What was the ranges HIDI was tuned over?
3. Were all the baselines HIDI was compared against tuned? If so, over what ranges?
4. Line 204: It is unclear how the objective in equation (7) mitigates non-stationarity in hierarchical models.
5. All environments were single-task environments. Shouldn’t the usefulness of subgoals be tested on multi-task environments, where the re-usuability of subgoals comes into play.
6. Line 313: The GP’s mean is described as “promising subgoal”. If this is promising for the main task, why then is it used as an exploration subgoal? It would be helpful if the meaning of “promising” could be clarified, and if the paper could describe how the mean of the GP’s predictive distribution would be promising in terms of exploration.
7. Why were the ablations (Fig 3 and 4) only conducted on Ant-Maze
8. Is the Baseline curve in Fig 3 (left) effectively HIRO? If so, it would be useful to mention that in section 4.2 under “How do various design choices within HIDI impact its empirical performance and effectiveness?”

---

### Official Review · Reviewer_kYYy · 2024-11-13

**Soundness:** 3
**Presentation:** 3
**Contribution:** 2
**Rating:** 5
**Confidence:** 3

**Summary:**

The paper proposes using diffusion model-based subgoal generation to solve the non-stationarity issue in HRL. It proposes using Gaussian priors for quantifying the uncertainty, and inducing states for exploration and compares their proposed algorithm with baselines in HRL.

**Strengths:**

* The use of diffusion model and GPs, and incorporating them with baselines in HRL for better performance
* Thorough ablation study on the three components of the proposed new algorithm
* Although the new proposed algorithm comes with new hyperparameters, the paper specifies the decision on they have chosen them
* Comprehensive comparison of the proposed algorithm with other baselines in HRL across multiple environments
* The paper provides mostly sufficient details about the computational complexity and reproducibility for the proposed algorithm

**Weaknesses:**

* It is not clear how the proposed algorithm helps with non-stationarity issues, and how this issue is disentangled from other factors in HRL.
* It is stated in the paper that HIDI is more sample efficient, however, sample efficiency is not directly measured in the experiments.
* More analysis of the components of HIDI including the hyperparameters would be beneficial
* Not enough reproducibility details for other baselines


Minor fixes include:

* Line 527 should be Figure 5, not 4.
* Figure 3, should mention the type of shaded regions and the number of seeds
* Experiments and related work should be separated

**Questions:**

1. Figure 5 is presented to demonstrate the improvement HIDI brings in comparison with HESS, due to its exploration strategy. How is this plot different from Figure 2, aside from including other algorithms along these two, and if it's the same, why are performance lines different ( HESS reaches a success rate of 0.2 in Figure 5, but a success rate of 0.6 in Figure 2, in Reacher)

2. The performances of all the baselines are too close in the dense reward configuration. How do you disentangle the effect of sparse reward, and the non-stationarity, and how does your proposed algorithm cause improvement?

3. The paper mentions that HIDI has more sample efficiency. How do you demonstrate that? It would be beneficial to define a measure of sample efficiency e.g. number of samples trained to reach a target performance, to compare all the baselines more precisely on this metric, and not on the performance plot only.

4. What is the reasoning behind using the RBF kernel aside from the fact that it is a common kernel for GPs? Since part of the contribution of the paper’s algorithm is using GPs, it would be beneficial to experiment on other kernels too, based on the assumptions about data

5. In Table 3, the paper mentions the hyperparameters of HIDI. What about other baselines?  it would be beneficial to mention how hyperparameters were dealt with for other baselines, and across environments, and if any tuning was done, to make a more fair comparison

6. Different values for the new hyperparameters of HIDI were only compared on Ant Maze. How are the values for them going to be chosen for other environments in general, when applying HIDI to a new environment?

---

### Note · Authors · 2024-12-03

**Comment:**

We sincerely thank the reviewers for their valuable and constructive feedback. Due to limited time to complete the recommended experiments, we have decided to withdraw our submission. We deeply appreciate the insightful comments and will work to address them thoroughly. Thank you again for your time and effort in reviewing our work.

**Withdrawal Confirmation:**

I have read and agree with the venue's withdrawal policy on behalf of myself and my co-authors.